# Melatonin-Nitric Oxide Crosstalk and Their Roles in the Redox Network in Plants

**DOI:** 10.3390/ijms20246200

**Published:** 2019-12-09

**Authors:** Ying Zhu, Hang Gao, Mengxin Lu, Chengying Hao, Zuoqian Pu, Miaojie Guo, Dairu Hou, Li-Yu Chen, Xuan Huang

**Affiliations:** 1Provincial Key Laboratory of Biotechnology of Shaanxi, Key Laboratory of Resource Biology and Biotechnology in Western China, Ministry of Education, College of Life Science, Northwest University, Xi’an 710069, China; 201731793@stumail.nwu.edu.cn (Y.Z.); gaohang19861222@163.com (H.G.); lumengxin666@163.com (M.L.); haocy@stumail.nwu.edu.cn (C.H.); 201820907@stumail.nwu.edu.cn (Z.P.); 201920904@stumail.nwu.edu.cn (M.G.); dairuhou@stumail.nwu.edu.cn (D.H.); 2Fujian Provincial Key Laboratory of Haixia Applied Plant Systems Biology, Center for Genomics and Biotechnology, Fujian Agriculture and Forestry University, Fuzhou 350002, China

**Keywords:** melatonin, nitric oxide, abiotic stress, biotic stress, reactive oxygen species, plant growth and development

## Abstract

Melatonin, an amine hormone highly conserved during evolution, has a wide range of physiological functions in animals and plants. It is involved in plant growth, development, maturation, and aging, and also helps ameliorate various types of abiotic and biotic stresses, including salt, drought, heavy metals, and pathogens. Melatonin-related growth and defense responses of plants are complex, and involve many signaling molecules. Among these, the most important one is nitric oxide (NO), a freely diffusing amphiphilic biomolecule that can easily cross the cell membrane, produce rapid signal responses, and participate in a wide variety of physiological reactions. NO-induced S-nitrosylation is also involved in plant defense responses. NO interacts with melatonin as a long-range signaling molecule, and helps regulate plant growth and maintain oxidative homeostasis. Exposure of plants to abiotic stresses causes the increase of endogenous melatonin levels, with the consequent up-regulation of melatonin synthesis genes, and further increase of melatonin content. The application of exogenous melatonin causes an increase in endogenous NO and up-regulation of defense-related transcription factors, resulting in enhanced stress resistance. When plants are infected by pathogenic bacteria, NO acts as a downstream signal to lead to increased melatonin levels, which in turn induces the mitogen-activated protein kinase (MAPK) cascade and associated defense responses. The application of exogenous melatonin can also promote sugar and glycerol production, leading to increased levels of salicylic acid and NO. Melatonin and NO in plants can function cooperatively to promote lateral root growth, delay aging, and ameliorate iron deficiency. Further studies are needed to clarify certain aspects of the melatonin/NO relationship in plant physiology.

## 1. Introduction

Melatonin (N-acetyl-5-methoxytryptamine), a tryptophan derived biomolecule, was initially discovered in the bovine pineal gland [1]. In animals, melatonin has been found to affect circadian rhythms, body temperature, mood, sleep, locomotor activity, food intake, sexual behavior, and immune responses [2]. Melatonin was not conclusively identified in higher plants until 1995 and animal melatonin has received far more research attention than plant melatonin [3]. The studies to date indicate that melatonin in plants is involved in growth regulation, including delaying leaf senescence [4], promoting root regeneration [5], embryo morphogenesis [6], regulating circadian rhythm, photoperiod, and flowering [7,8] resistance to pathogen invasion [9], and ameliorating the damage caused by substances such as heavy metals [6], salt ions, and other chemicals [10], UV radiation [11], and temperature changes [12]. Melatonin is considered to function primarily as an antioxidant, particularly in the control of reactive oxygen species (ROS), reactive nitrogen species (RNS), and other free radicals and harmful oxidative molecules in plant cells [5].

The effects of plant melatonin are associated with feedback mechanisms of various regulatory elements of the redox network, including ROS and RNS, particularly through the action of nitric oxide (NO) [13]. However, excessive RNS and especially ROS accumulation can lead to oxidative damage in cells. The regulation of the redox homeostasis balance has become a very important research area. NO, an important signaling molecule, has both pro-oxidant and antioxidant properties. The dual effects of NO mainly depend on its local concentration and spatial generation mode. In response to the exposure of plants to abiotic and biotic stresses, NO may react with other redox-related molecules and may regulate protein function through a variety of different mechanisms [14]. In addition, NO is involved in plant growth and development, seed dormancy and germination, root morphogenesis, and the regulation of stomatal movement [15].

It has been shown that melatonin may reduce oxidative damage, by inhibiting NO production and the activity of NO synthase [16]. In addition, when 5-hydroxytryptophan derivatives like serotonin and N-nitroso-melatonin (NOMela) were reacted with each other, it was shown that NOMela releases NO [17]. Melatonin regulates the NO/NOS (nitric oxide synthase) system through a variety of mechanisms that may affect physiological and pathophysiological processes [18]. Melatonin has also been reported to play an important role in plant stress responses through its interaction with NO. NO may act downstream of melatonin to promote plant defenses against a variety of biotic and abiotic stresses [19]. Melatonin promotes the accumulation of NO by triggering the activation of the arginine pathway [20]. It also increases the activity of NO synthase by up-regulating the expression of related genes, leading to an increase in NO content and consequent effects on plant growth and development, e.g., delayed fruit senescence [21]. The interaction between NO and melatonin exhibits a certain degree of complexity due to the fact that they interact independently and through multiple signaling pathways. Here, we review the recent literatures updating the relationships between melatonin and NO in plants (summarized in Table 1), and give perspectives for future research.

## 2. Research Progresses on Melatonin in Plants

The biosynthesis of melatonin in animals has been extensively studied in the past decades and is reasonably well understood. In contrast, studies of melatonin biosynthesis in plants are recent (mostly since 2012), and our knowledge is fragmentary. The complexity of the synthetic pathway is reflected by the discovery of five enzymes involved in the overall conversion of tryptophan to melatonin—tryptophan decarboxylase (TDC) [40], tryptamine 5-hydroxylase (T5H) [41], serotonin N-acetyltransferase (SNAT) [42], N-acetylserotonin methyltransferase (ASMT) [43], and caffeic acid O-methyltransferase (COMT) [42]. In 2000, Saxena’s group was the first to discover, using a radioisotope tracer method, that tryptophan is the main precursor of melatonin biosynthesis in plants [44]. Tryptamine is first generated from tryptophan under TDC catalysis; then serotonin is generated from tryptamine under T5H catalysis of T5H, N-acetylserotonin is generated from serotonin under SNAT catalysis, and melatonin is generated from N-acetylserotonin under ASMT catalysis. An enzyme involved in phenylpropanoid metabolism, COMT, may substitute for ASMT in the conversion of N-acetylserotonin to melatonin. This is the primary pathway for melatonin synthesis. However, there is an alternative pathway in which 5-methoxytryptamine is generated from serotonin under ASMT or COMT catalysis, and melatonin is generated from methoxytryptamine under SNAT catalysis. The synthesis site of serotonin in the first two steps of melatonin biosynthesis is in the endoplasmic reticulum. The synthesis site of melatonin is either in the (i) cytoplasm if the last-step synthetase is ASMT or COMT, or (ii) chloroplast if the last-step synthetase is SNAT [45].

Melatonin functions mainly as an antioxidant, because it is soluble in both water and fat, and can move freely in any moist areas of the body [46]. Melatonin in plants is considered to be a phytohormone, structurally similar to auxin, which also uses tryptophan as the precursor substance. Similarly to auxin, melatonin promotes plant growth and development. It can increase the root growth rate of Phalaris, Triticum, and Arabidopsis 3- to 4- fold relative to the control plants [47]. Melatonin can also delay the aging process of plants. Leaves treated with various concentrations of cytokinin had a similar phenotype to those treated with melatonin, suggesting that these substances play a synergistic role in delaying plant senescence [48]. Melatonin may interact with the phytohormones gibberellin (GA) or abscisic acid (ABA) during plant growth and development. The exogenous melatonin treatment of cucumber seedlings under saline conditions up-regulated the expression of GA and GA biosynthesis genes (*GA20ox* and *GA3ox*), and down-regulated the expression of ABA and ABA biosynthesis genes (*LpZEP* and *LpNCED1*) [10]. Plants are frequently attacked by various pathogens during growth. Melatonin acts upstream of the signaling pathways of defensive substances, including salicylic acid (SA), jasmonic acid (JA), NO, and ethylene, which work together to enhance plant disease resistance [49]. In a recent study, exogenous melatonin promoted anthocyanin accumulation in crabapple leaves, and increased flavonol and proanthocyanin levels by eliminating the need to rely on light [50].

The receptor for melatonin in plants has not been conclusively identified. Therefore, despite recognition of melatonin as a phytohormone, its precise function and signaling pathways remain unclear. Q. Chen’s group recently reported the first putative plant melatonin receptor (candidate G protein-coupled receptor 2/phytomelatonin receptor, CAND2/PMTR1) in Arabidopsis, and showed that melatonin controls receptor-dependent stomatal closure [51]. CAND2 is a membrane protein that interacts with the G-protein α-subunit (GPA1), and the expression of AtCand2 is regulated by melatonin in various organs and guard cells. The regulation of stomatal closure by plant melatonin is evidently based on CAND2/PMTR1-mediated hydrogen peroxide (H_2_O_2_) and the Ca^2+^ signaling cascade. This section may be divided by subheadings. It should provide a concise and precise description of the experimental results, their interpretation as well as the experimental conclusions that can be drawn.

## 3. Nitric Oxide Synthesis and Signaling Pathways in Plants

NO was proved to be an important signaling molecule (the first gas signaling molecule) and effector molecule in a wide variety of organisms in the late 1980s. It is able to easily enter cells to directly activate effector enzymes and participate in numerous physiological and pathological processes. In mammals, three different NO synthase (NOS) enzymes have been discovered that oxidize arginine to produce NO [52,53]. The quantification of NO in plant material is difficult, and it is also impossible to ascertain its subcellular localization; therefore, NO production in plants is a subject of controversy [54]. Sources of NO in plants can generally be classified as enzymatic or non-enzymatic. One of the major enzymatic reactions is catalyzed by nitrate reductase (NR). Nitrate (NO^3−^) or nitrite (NO^2−^) is generated through NR activity. Nitrite is then reduced to NO and the derivative peroxynitrite (ONOO^−^) by NR itself or by the mitochondrial electron transport chain [55]. Nitrate metabolism is considered as the first system for NO production [56]. Another enzyme active under hypoxic conditions is xanthine oxidoreductase (XOR). Cytochrome P450 (also termed copper amine oxidase 1) is a potential source of NO [57]. Despite early reports of NOS isolation from higher plants, NOS homologs have been identified only in algae [58]. There is no clear evidence for the presence of NOS-like proteins in plants [59]. Non-enzymatic sources of NO include carotenoids, phenolic compounds, and ascorbic acid [60,61,62]. NO can be generated non-enzymatically from nitrite under acidic conditions [63].

NO molecules play important roles in post-translational modifications (PTMs) of proteins. The most common NO-mediated PTM is S-nitrosylation, a reversible redox modification based on the addition of a nitroso group to a thiol group (SH) of a specific cysteine (Cys) residue, and the consequent production of S-nitrosothiol (SNO), which can cause a conformational change and altered activity or function of a protein [64,65]. The second most common NO-mediated PTM is tyrosine nitration [66]. Nitrification of tyrosine alters superoxide dismutase (SOD) activity, with the consequent change of ROS signaling, peroxynitrite (ONOO^−^) production, and tyrosine nitration of protein [67].

The precise function of NO in cells remains unclear, but numerous studies indicate that NO production is a step in the signal cascade involving ROS, phytohormones, protein kinases, and second messengers such as Ca^2+^ and cyclic guanosinc monophosphate (cGMP). The two pathways that mediate signal transduction of NO can generally be classified as cGMP-dependent and cGMP-independent. As the second messenger of the cGMP signaling pathway, NO plays a crucial role in root growth and development [68]. NO can activate the MAPK cascade; e.g., 46-kD MAPK in maize (*Zea may*s) [69]. MAPK signalling regulates NO and nicotinamide adenine dinucleotide phosphate (NADPH) oxidase-dependent oxidative burst in *Nicotiana benthamiana* [70]. In Arabidopsis, NR is encoded by two genes, NIA1 and NIA2. NIA2, an NR isoform, interacts directly with mitogen-activated protein kinase 6 (MPK6) (MPK6 from *Arabidopsis thaliana*) in vitro and in vivo, and MPK6 can phosphorylate NIA2. Phosphorylated NIA2 significantly increased NIA2 activity and NO production, and also caused changes of Arabidopsis root morphology [71]. MPK1/2 is involved in the induction of NR-dependent NO production, and the accumulation of nitrosoglutathione in NO-derived RNS may lead to S-nitration of NR [72].

NO and Ca^2+^ both function as second signals in plant cells, and NO production is dependent on the presence of Ca^2+^ [73]. Innate immune responses in plants, involve the increase of intracellular Ca^2+^ levels, with the consequent activation of calmodulin (CaM) or CaM-like proteins, and the promotion of NO biosynthesis [74]. Increased Ca^2+^ level stimulates the production of SA, NO, and ROS, triggering programmed cell death in areas of infection, and limiting the growth of pathogens. CaM and CaM-like proteins, the downstream targets of Ca^2+^ sensors and Ca^2+^ signaling, respectively, were reported to be involved in the pathogen-associated molecular pattern (PAMP)-induced NO synthesis [73]. NO can affect the Ca^2+^ concentration in cytosol by adjusting the Ca^2+^ channel “switch” on the plasma membrane. The production of H_2_O_2_, Ca^2+^, and NO in *Arabidopsis* guard cells promotes stomatal closure [75].

The production of NO and ROS in plant cells can also be induced by hormones or environmental stresses. NO and ROS interact with H_2_O_2_ and are involved in stress resistance. ROS are most likely located upstream of NO. The dynamic balance of ROS and NO is a key factor in determining the redox state of plants, and is essential for normal plant activities [76]. In tobacco (*Nicotiana tabacum*), NO and H_2_O_2_ jointly regulate brassinosteroid (BR)-mediated antiviral responses, and NR-mediated NO production is dependent on the respiratory burst oxidase homolog (RBOH)-mediated H_2_O_2_ production [77]. A research reported that under stress conditions, the ability of NO to scavenge H_2_O_2_ was enhanced by the thiol nitrosylation of the 32nd Cys of cytosolic ascorbate peroxidase1 (APX1). A direct link was established between NO and H_2_O_2_ signaling pathways, demonstrating the regulation of redox balance by NO [78].

## 4. Regulatory Roles of Melatonin and Nitric Oxide in Stress Tolerance in Plants

During their growth and development, plants are subjected to a variety of biotic and abiotic stresses, which can potentially lead to growth inhibition, aging, reduced production, or even death. Melatonin, and its precursors and derivatives, play significant roles as growth regulators, biostimulants, and antioxidants under abiotic and biotic stress conditions, by directly scavenging ROS, and delaying leaf senescence, reversing inhibition of photosynthesis, and maintaining the redox and RNS steady state [79]. Exogenous melatonin promotes plant growth, photosynthesis, and antioxidant activity, and enhances the tolerance of drought, extreme temperatures, high salinity, heavy metals, acid rain, and pathogens. Melatonin is involved in the improvement of many physiological processes. In apples (*Malus prunifolia*), melatonin prolonged the normal growth period and protected the new tissue from environmental injury and stresses [80]. The application of exogenous melatonin under stress conditions leads to increased endogenous melatonin and NO production. The relationships (crosstalk) between melatonin and NO under abiotic and biotic stresses are discussed in Section 4.1 and Section 4.2 below.

### 4.1. Abiotic Stresses

Abiotic stresses will seriously impact plant growth and reproduction, and reduce the yield of most crop species. Plants have evolved several complex mechanisms to adapt to environmental stresses. In response to abiotic stress, melatonin acts as a potent antioxidant that up-regulates the transcription levels of several antioxidant enzymes and increases SOD, catalase, glutathione peroxidase (GPx), glutathione reductase (GR), and glucose levels [81]. Antioxidant enzymes such as glucose -6-phosphate dehydrogenase (G6PD) enhances stress resistance [10]. Melatonin can also enhance resistance by altering the Na^+^/K^+^ ratio [22]. NO and melatonin show a similar mode of action in plant cells. NO is a key factor in the abiotic stress response. It reacts with ROS to improve redox homeostasis, promote tolerance of oxidative stress, and enhance antioxidant capacity [82]. NO can interact directly or indirectly with a wide variety of targets, with the consequent modulation of protein function and reprogramming of gene expression. Studies on the relationships between melatonin and NO under specific types of abiotic stress are described in the subsections below and summarized in Figure 1.

#### 4.1.1. Salt Stress

Salt stress is one of the major limiting factors in agricultural production worldwide, and is responsible for huge yield losses each year [83]. High salinity causes osmotic stress, and excessive Na^+^ accumulation results in high levels of ROS [84]. The application of exogenous melatonin and consequent increase in endogenous melatonin, enhanced salt stress resistance in *Arabidopsis*, *Malus hupehensis*, and maize seedlings [85,86,87]. In cucumber (*Cucumis sativus*) seedlings under salt stress, melatonin treatment improved the net photosynthetic rate, photosystem II (PSII) maximum quantum efficiency (Fv/Fm), and total chlorophyll content [10]. Melatonin-induced salinity tolerance participates in the ROS signaling route. The respiratory burst oxidase protein F (*AtrbohF*), an *Arabidopsis rbohF* mutant, reestablishes redox and ion homeostasis by regulating ROS production in salinity tolerance triggered by melatonin [85]. ROS are most likely located upstream of NO. Therefore, these studies have shown that melatonin/NO regulate the production of ROS and oxidative defence signals in plants.

Melatonin and NO-releasing compounds can modulate the transcription of the sodium hydrogen exchanger (NHX1) and salt overly sensitive 2 (*SOS2*), thereby maintaining the Na^+^/K^+^ ratio [29]. Glutathione in plants is an important antioxidant whose electron-donating ability and high intracellular concentration help maintain proper cellular environment [88]. The ratio of reduced/ oxidized glutathione (GSH/GSSG), an indicator of cellular oxidative status, is regulated by glutathione reductase (GR) [81]. In sunflower seedling cotyledons, melatonin and NO differentially alleviated salt stress by regulating GR activity and GSH content [24]. Sodium nitroprusside (SNP), acting as an NO source, promotes plant growth, has protective effects against salt stress, oxidation, and aging, and helps maintain ion homeostasis [89,90]. SNP reduced the activity of hydroxyindole-O-methyltransferase (*HIOMT*; a melatonin synthase) in the control sunflower seedling cotyledons, and was up-regulated in salt-stressed seedlings. The application of exogenous melatonin (15 μM) with or without NaCl (120 mM) inhibited seedling growth, and this effect was associated with NO availability, accumulation of superoxide anion (O^2−^) and peroxynitrite (ONOO^−^) anions, degree of tyrosine nitration of proteins, spatial localization, and activity of SOD isoforms. Under salt stress, the NO interaction with melatonin may act as a long-distance signal regulating seedling growth and maintaining oxidative homeostasis, in association with the differential regulation of two SOD isoforms, (Cu/ Zn SOD and Mn SOD) [25]. There is clearly a functional relationship between melatonin and NO during salt stress. There is some evidence that NO-induced S-nitrosylation is involved in salt stress responses. The removal of NO did not affect endogenous melatonin content in roots added with NaCl or with melatonin. The inhibition of nitrate reductase activity in mutants (*nia1/2*, *noa1*) caused an indirect reduction in endogenous NO levels which was not restored by melatonin supplementation [29]. NO evidently acts downstream of melatonin in promoting salt tolerance. In pepper (*Capsicum annuum*), salt stress was alleviated by spraying melatonin, which led to a further increase in NO and H_2_S levels. The melatonin-induced tolerance of salt stress was suggested to be related to the downstream signal crosstalk between NO and H_2_S [22].

#### 4.1.2. Heavy Metals

Heavy metals are metal elements that have a density higher than that of water [91]. In contrast to organic pollutants, they cannot be degraded or converted by microorganisms. Soils contaminated by heavy metals are typically, difficult to remediate. In plants, heavy metals cause oxidative stress by altering the content or activity of antioxidant enzymes [92]. Similarly to ROS, RNS (particularly NO) triggers the signaling pathway-induced expression of related resistance genes [93]. A relationship exists between NO and melatonin (see preceding section), and melatonin may promote resistance to heavy metal stress in some cases.

Studies during the past decade have addressed the effects of melatonin on plants under cadmium (Cd) stress [86,94]. Cd stress may induce the transcription of melatonin biosynthesis genes, and consequent melatonin production [95]. Such Cd-induced melatonin synthesis is dependent on light, H_2_O_2_, and NO. Cd-stressed rice cells show increased levels of H_2_O_2_ and NO, which activate the MAPK cascade reaction. Yet-unidentified transcription factors are then up-regulated or down-regulated, leading to the induction of melatonin biosynthesis genes (e.g., *TDC*, *T5H*) or inhibition of *SNAT* and *COMT*, and consequent increase of melatonin content [23]. In Cd-stressed wheat seedlings, melatonin enhanced the tolerance of Cd toxicity and antioxidant defense mechanisms, by increasing endogenous NO [28]. These findings suggest that the crosstalk between melatonin and NO facilitates the Cd stress response.

High concentrations of lead (Pb) in plants typically cause inimical changes in cell morphology, reduce cell growth, and eventually lead to cell death [96]. In maize plants exposed to Pb for 10 days, melatonin (0.05 or 0.10 mM) treatment resulted in reduced Pb toxicity and enhanced endogenous NO content. cPTIO, a NO scavenger, abolished the melatonin-induced Pb toxicity tolerance by reducing endogenous NO. These findings indicate that NO is involved in melatonin-induced antioxidant defense, and melatonin/ NO interaction plays an important role in Pb stress tolerance in maize [31]. In addition, these results provide a new perspective for the study of melatonin signal transduction and the complex molecular system in plants under Pb toxicity.

Aluminum (Al) toxicity is one of the major limiting factors for plant growth in acidic soils (pH <5.5), in which Al complexes in aluminum silicate clay dissolve into the most toxic trivalent cation, Al^3+^. Al^3+^ mainly poisons root tips and inhibits root growth [97]. In plant roots, NO may alleviate oxidative stress and cell wall polysaccharides under Al toxicity [98]. Al stress may reduce the expression of *SNAT*, resulting in decreased endogenous melatonin [99]. Under Al-stressed *Arabidopsis*, *snat* mutants displayed enhanced stress sensitivity, due to the NR- and NOS-dependent NO production, and suppressed root growth, compared with the wild type. The application of p-chlorophenylalanine (p-CPA), a melatonin synthesis inhibitor, leads to an increase in NO content under Al stress [32]. Melatonin may interfere with NO-mediated cell division cycle progression and the quiescent center activity, alleviating Al-induced root growth inhibition.

#### 4.1.3. Drought Stress

Damage caused by drought stress is alleviated by the exogenous application of melatonin in certain species, including soybean, *Medicago truncatula*, and cucumber [30,100,101]. iTRAQ proteomic analyses indicated that melatonin-treated corn, relative to non-treated controls, had higher levels of carbon fixation, photosynthesis, biosynthesis of amino acids, and secondary metabolites, and differentially expressed proteins (DEPs). Melatonin treatment reduced cell membrane damage by significantly increasing the photosynthetic rate, cell turgor, and water-retention capacity. Long-term melatonin treatment inhibited the expression of aging-related genes (*SAG12*, *PaO*) and slowed down the drought-induced leaf senescence process [27]. Drought-stressed alfalfa (*Medicago sativa*) treated with melatonin had reduced levels of H_2_O_2_, and NO improved the enzymatic regulation of ROS and RNS. The changes in enzymatic and transcriptional levels helped maintain nitro-oxidative homeostasis [102]. However, how endogenous melatonin interacts with NO in drought stress is still poorly understood.

#### 4.1.4. Other Stresses

Melatonin and NO have been found to alleviate plant damage by reducing Na^+^ accumulation, activating genes involved in defense response signaling pathways, and increasing K^+^ uptake, antioxidant enzyme activity, and ASA glutathione detoxification. The alleviation of sodic alkaline stress by exogenous melatonin evidently requires NO as a downstream signal [30]. Under alkaline stress, melatonin up-regulates the transcription of the Na extraction gene (*SlSOS1*) in the SOS (salt overly sensitive) signaling pathway, thereby opening up a new signaling pathway for improved plant resistance.

The transcription factors of the *Cysteine 2/Histidine 2* (*C_2_H_2_*) and *Zinc Finger* (*ZATs*) genes are sometimes essential for inducing tolerance to biotic and abiotic stresses. In tomato fruit, the induction by exogenous melatonin enhanced cold tolerance by up-regulating *ZAT2*/6/12, and promoted NO accumulation by triggering the arginine pathway activity [20].

Exogenous melatonin improved plant tolerance to photo-oxidative stress in a hydrogen peroxide (H_2_O_2_)-dependent manner. NO interacts with H_2_O_2_ and is involved in many aspects of plant metabolism, including stress resistance. The resistance of the chlorophyte (green algae) *Haematococcus pluvialis* to nitrogen starvation or light stress was improved by exogenous melatonin. Melatonin-induced the accumulation of the *H. pluvialis* activated NO-dependent MAPK signal cascade, indicating that MAPK is a target of NO action in physiological processes [33]. The physiological pathways involving stress-related transcription factors remain to be further studied.

### 4.2. Biotic Stresses

In natural environments, plants routinely encounter damaging organisms, including bacteria, viruses, fungi, and herbivores. Many studies in the past decade have focused on pathogenic bacteria, and the multiple layers of defense that plants have developed against infection [103]. Ma’s group discovered that the exogenous application of melatonin improved the resistance of *Malus prunifolia* to *Marssonina* apple blotch [80]. In a *Arabidopsis snat* mutant, avirulent pathogen *Pseudomonas syringae pv.* down-regulated the expression of defense genes (*PR1*, *ICS1,* and *PDF1.2*) [104]. The exogenous application of melatonin enhanced plant resistance to pathogens, such as *Fusarium oxysporum*, *Penicillium spp.*, *Phytophthora infestans*, *Botrytis cinerea*, and *Rhizopus stolonifer* [79]. In cotton, the exogenous application of melatonin enhanced the expression of genes involved in the phenylpropanoid, mevalonate (MVA), and gossypol pathways following *Verticillium* dahliae inoculation [105]. NO, a signaling molecule upstream of the innate immune system in plants, plays a key role in the responses to pathogen invasion. Many studies have addressed NO-mediated plant disease responses, and the relationship between NO and SA [14]. SA-deficient (NahG-overexpressing) plants and NO-deficient mutants (*noa1* and *nia1nia2*) have a high sensitivity to bacterial pathogens. Both SA and NO enhanced the resistance of *Arabidopsis* to pathogenic bacteria, and the synergy between the two compounds played an important role in natural immunity [36]. *Arabidopsis* treated with melatonin followed by *Pseudomonas syringe pv. tomato* (*Pst*) DC3000 infection rapidly increased the NO level in leaves, and the NO donor and NO scavenger were present endogenously [26]. Endogenous melatonin production did not play a significant role in the process. These findings indicate that NO acts as a downstream signal for melatonin in the plant immune response. Increased NO levels was associated with increased expression levels of SA synthetic genes (*AtEDS1*, *AtPAD4*) and SA downstream genes (*AtPR1*, *AtPR2*, and *AtPR5*) [36]. After infection of *Arabidopsis* with *Pst* DC3000, melatonin treatment can induce transcriptional levels of *CBF/DREB1s*, thereby enhancing the accumulation of soluble sugars [34]. Comparative metabolomics analysis showed that the treatment of *Arabidopsis* with melatonin and *Pst* DC3000, led to increased levels of endogenous soluble sugars and glycerol, which consequently increased the levels of SA and NO, and activation of immune responses to pathogens [35].

Infection by pathogenic bacteria may also cause increased levels of H_2_O_2_ and NO, leading to higher endogenous melatonin level activation of the MAPK cascade via oxidative signal-inducible1/MAPKK kinases 3–MAPK kinases4/5/7/9-MAPK3/6 (OXI1/MAPKKK3-MAPKK4/5/7/9-MAPK3/6). Such MAPK cascade activation resulted in increased SA levels, which induced the expression of several defense-related genes, including *PR1* [37]. These findings, taken together, suggest that the melatonin-mediated pathogen invasion is coupled with an NO-mediated plant defense-signaling pathway, as shown schematically in Figure 2. Whether there is a certain relationship between melatonin/NO and other plant hormones, e.g., JA and piperidine acid, is unknown. Further studies on the relationship between melatonin and NO are needed in the direction of genomics, proteomics, and transcriptomics.

## 5. Regulatory Roles of Melatonin and Nitric Oxide in Plant Growth and Development

Melatonin has many essential functions during plant growth and development. Melatonin appears to be related to auxin in view of its chemical structure and biosynthetic pathway, and some studies have addressed the effects of exogenous melatonin on root development [106]. Research showed that the melatonin treatment of *A. thaliana* resulted in a 2-fold increase in adventitious root formation (ARF) and 3-fold increase in lateral root formation [107]. Root growth may be affected differently depending on the melatonin concentration; *e.g.*, treatment of sweet cherry with 1 μM of melatonin increased the number and length of roots, whereas 10 μM of melatonin inhibited root growth [108]. In studies of tomato seedlings, NO may participate in melatonin-induced ARF as a downstream signal. The exogenous application of melatonin promoted or inhibited ARF depending on with concentration, and caused an accumulation of endogenous NO by down-regulating the expression of S-nitrosoglutathione reductase (*GSNOR*). Melatonin affects auxin transport, signal transduction (*PIN1*, *PIN3*, *PIN7*, *IAA19*, and *IAA24*), and auxin accumulation through the NO signaling pathway [39]. The overlapping roles of melatonin, NO, and auxin in plant root development and signal transduction were further understood.

Melatonin can affect the vegetative growth stage as well as the reproductive growth stage of plants. Because melatonin delays senescence in various fruit species, it is widely used in postharvest storage [109,110]. The proteomics study showed that of tomato fruit treated with 50 μM of melatonin, significantly affected 241 proteins, including those involved in cell wall formation, oxidative phosphorylation, carbohydrate metabolism, and fatty acid metabolism. The exogenous application of melatonin also promoted anthocyanin accumulation during the maturation process [111]. Similarly, NO regulated the senescence of mature fruits; i.e., it delayed ripening by inhibiting ethylene biosynthesis and preventing downstream reactions [112]. Melatonin and NO blocked the up-regulation of polygalacturonase (PG)- and cellulase (Cel)-related genes (*PcCe*l, *PcPG*), inhibited the expression of ethylene synthase genes (*PcACS*, and *PcACO*), and reduced respiratory and ethylene production. Melatonin enhanced NOS activity by up-regulating *PcNOS* expression, with a consequent increase of NO content, inhibition of ethylene production, and delay of fruit senescence [21]. The NO synthetic pathway for melatonin-induced NR/Ni-NOR remains to be elucidated.

Iron (Fe), an important nutrient element during plant growth, plays key roles in photosynthesis, DNA synthesis, respiration, and hormone synthesis [113]. The total iron content is fairly high in most soils; however, the content of soluble iron that can be absorbed and utilized by plants is low. Iron deficiency often causes chlorosis and inhibits normal plant growth [114]. The application of exogenous polyamines causes an increase in NO production, indicating that NO is a key intermediate in the polyamine-mediated signaling pathway [38]. NO can increase the soluble iron content, and ameliorate leaf chlorosis and oxidative stress. Under iron-deficient conditions, elevated melatonin levels lead to an increased polyamine content, which induces NO accumulation. The NO signal up-regulates the expression of iron-related genes such as *FIT1*, *FRO2*, and *IRT1*, thereby inducing the reactivation of iron in the cell wall, and increased availability of soluble iron. These phenomena were completely suppressed in the polyamine- and NO-deficient plants [38]. The NO-melatonin crosstalk is therefore partially related to the polyamine-mediated signaling pathway. However, the study of melatonin signaling in plants is still in its infancy, and further studies are needed to elucidate the role of melatonin signaling pathways in plants.

## 6. Conclusions

NO, a gas signaling molecule present in a wide variety of organisms, has numerous physiological functions. It helps maintain normal growth and development of plants, and also plays essential roles in responses to various biotic and abiotic stresses. Melatonin is another “hot topic” in plant research during the past two decades. NO and melatonin act as scavengers for active oxygen free radicals, and are involved in plant growth and development. Many studies have addressed the relationship between NO and melatonin, but there are still major gaps in our knowledge. The S-nitrosation target proteins have not yet been identified, and the NO signaling pathway is not well understood. Relatively few studies have focused on the receptors for plant melatonin, or the intermediate compounds in the melatonin metabolism pathway. Most of the studies on melatonin and NO are still focused on exogenous melatonin, and there are few studies on endogenous melatonin, which is not convincing enough. Whether melatonin is involved in the signal directly mediated by NO, and what kind of relationship it has with other genes or other substances in the signalling pathways, still needed to be investigated. Future studies along with these lines will unveil the relationships between melatonin and NO signaling pathways.

## Figures and Tables

**Figure 1 ijms-20-06200-f001:**
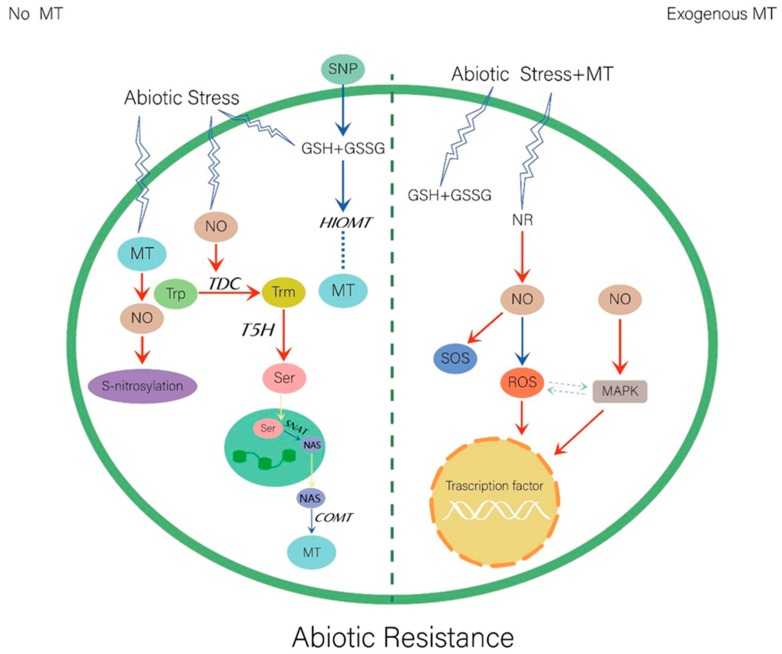
Relationship between the melatonin and nitric oxide in abiotic stress. MT, melatonin; NO, nitric oxide; SNP, sodium nitroprusside; Trp, tryptophan; Trm, tryptamine; Ser, sertonin; NAS, N-acetylserotonin; GSH, reduced glutathione; GSSG, oxidized glutathione; *TDC*, tryptophan decarboxylase; *TPH*, tryptophan hydroxylase; *T5H*, tryptamine 5- hydroxylase; *SNAT*, serotonin N-acetyltransferase; *COMT*, caffeic acid O-methyltransferase; NR, nitrate reductase; SOS, salt overly sensitive; *ROS*, reactive oxygen species; MAPK, mitogen-activated protein kinase cascade. The red arrows indicate increase; the blue arrows indicate decrease; the dotted lines indicate uncertainty.

**Figure 2 ijms-20-06200-f002:**
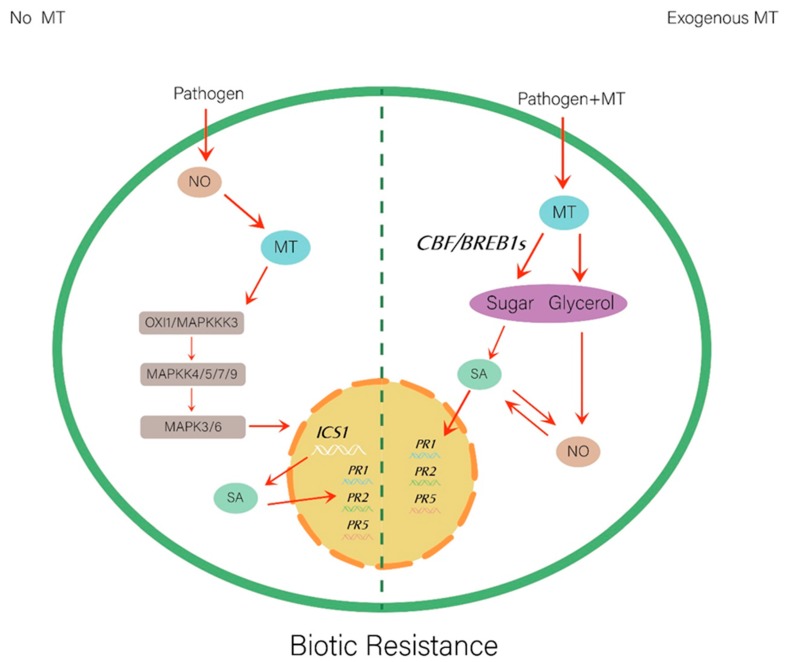
Relationship between the melatonin and nitric oxide in biotic stress. SA, salicylic acid; ICS1, isochorismate synthase 1; *PR1*, pathogenesis-related enzymes 1; *PR2*, pathogenesis-related enzymes 2; *PR5*, pathogenesis-related enzymes 5; *CBF/DREB1s*, C-repeat-binding factors/drought response element binding 1 factors; OXI1, oxidative signal-inducible 1.

**Table 1 ijms-20-06200-t001:** Relationship between the melatonin and nitric oxide in plants.

Plant	Source	Effect	Reference
*Capsicum annuum L.*	Exogenous melatonin	Iron deficiency and salt stress alone or in combination	[22]
Rice	Endogenous melatonin	Cadmium stress	[23]
Sunflower	Exogenous melatonin	Salt stress	[24,25]
*Arabidopsis thaliana*	Exogenous melatonin	Salt, drought, and cold stresses	[26]
*Medicago sativa*	Exogenous melatonin	Drought stress	[27]
Wheat	Exogenous melatonin	Cadmium stress	[28]
Rapeseed (*Brassica napus L.)*	Exogenous and endogenous melatonin	Salinity Stress	[29]
Tomato	Exogenous melatonin	Sodic alkaline stress	[30]
Maize plants	Exogenous melatonin	Pb stress	[31]
*Arabidopsis thaliana*	Exogenous and endogenous melatonin	Aluminum stress	[32]
Tomato	Exogenous melatonin	Chilling tolerance	[20]
*Haematococcus pluvialis*	Exogenous melatonin	High Light and Nitrogen Starvation Stress	[33]
*Arabidopsis thaliana*	Exogenous melatonin	*Pseudomonas syringe pv. tomato (Pst)* DC3000	[34,35]
*Arabidopsis thaliana*	Exogenous and endogenous melatonin	*Pseudomonas syringe pv. tomato (Pst)* DC3000	[36]
*Arabidopsis thaliana*	Endogenous melatonin	*Pseudomonas syringe pv. tomato (Pst)* DC3000	[37]
*Arabidopsis thaliana*	Exogenous melatonin	Iron Deficiency	[38]
*Solanum lycopersicum L.*	Exogenous melatonin	Root Development	[39]
*Pyrus communis L.*	Exogenous melatonin	Delay postharvest	[30]

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
