# Peer review of "Melatonin-Nitric Oxide Crosstalk and Their Roles in the Redox Network in Plants"

_ijms, 2019, doi:10.3390/ijms20246200_

Round 1

Reviewer 1 Report

Chapter „4. Crosstalk between NO and melatonin under stress conditions“ starts in line 175. On the other hand, you are providing important information in preceding chapters. Therefore my question is, whether the headline of your review is well chosen. - Throughout your text you are describing quite often the effects of NO and melatonin rather than a crosstalk between these two molecules.

You have to use quite a number of shortages. Each shortage is explained in the text. Nevertheless, I wonder whether shortages should be listed as well at the bottom of the first page of their respective use.

Author Response

 Response:

Chapter 4 has been changed to “Regulatory roles of melatonin and nitric oxide in stress tolerance in plants” in line 179. 4.1 has been changed to “Abiotic stresses” in line 202 . 4.2 has been changed to “Biotic stresses” in line 321.   Chapter 5 has been changed to “Regulatory roles of melatonin and nitric oxide in plant growth and development” in line 363. All of shortages have been explained as well at the bottom of the first page of their respective use in the text.

Reviewer 2 Report

In recent years, the role of melatonin in a multitude of physiological processes in plants has been clearly established, considering it with all justice a phytohormone with all the connotations that it has. The review proposed by the authors seems very appropriate. It is well organized, with an adequate bibliography and treats with a very good depth the most relevant aspects of the relationship of melatonin with NO and all types of stress in which it is related in some way. I think it is well written, although English is not my mother tongue, I cannot say it with total certainty.

            There is one aspect of the relationship between melatonin and NO that although it is implicit throughout the text, I have the feeling that it should have more prominence in the title and is related to the improvement of oxidative stress in all stress situations discussed. This is a key word, but I think it should also go in the title because the improvement of oxidative stress by modulating the free radical scavenging is a hotspot in this whole topic.

Author Response

Response:

The title has been changed to “Melatonin-nitric oxide crosstalk and their roles in redox network in plants”.